# Comparison of Ceftolozane/Tazobactam Infusion Regimens in a Hollow-Fiber Infection Model against Extensively Drug-Resistant *Pseudomonas aeruginosa* Isolates

María Milagro Montero,[a,b,c,d,h] Sandra Domene-Ochoa,[a,b,c,d] Carla López-Causapé,[e] Inmaculada López-Montesinos,[a,b,c,d] Sonia Luque,[f,h] Luisa Sorlí,[a,b,c,d,h] Núria Campillo,[f] Eduardo Padilla,[g] Núria Prim,[g] Lorena Ferrer Alapont,[a,b,c,d] Santiago Grau,[f,h] Antonio Oliver,[e,h] Juan P. Horcajada[a,b,c,d,h]

ªInfectious Diseases Service, Hospital del Mar, Barcelona, Spain
ᵇInfectious Pathology and Antimicrobials Research Group (IPAR), Institut Hospital del Mar d'Investigacions Mèdiques (IMIM), Barcelona, Spain
ᶜUniversitat Autònoma de Barcelona (UAB), Bellaterra, Spain
ᵈCEXS-Universitat Pompeu Fabra Barcelona, Barcelona, Spain
ᵉServicio de Microbiología y Unidad de Investigación, Hospital Son Espases, IdISBa, Palma de Mallorca, Spain
ᶠPharmacy Service, Hospital del Mar, Barcelona, Spain
ᵍLaboratori de Referència de Catalunya, Barcelona, Spain
ʰCIBER of Infectious Diseases (CIBERINFEC CB21/13/00002), Institute of Health Carlos III, Madrid, Spain

**ABSTRACT** The aim of this study was to compare the efficacy of intermittent (1-h), extended (4-h), and continuous ceftolozane-tazobactam (C/T) infusion against three extensively drug-resistant (XDR) sequence type (ST) 175 *P. aeruginosa* isolates with different susceptibilities to C/T (MIC = 2 to 16 mg/L) in a 7-day hollow-fiber infection model (HFIM). C/T in continuous infusion achieved the largest reduction in total number of bacterial colonies in the overall treatment arms for both C/T-susceptible and -resistant isolates. It was also the only regimen with bactericidal activity against all three isolates. These data suggest that continuous C/T infusion should be considered a potential treatment for infections caused by XDR *P. aeruginosa* isolates, including nonsusceptible ones. Proper use of C/T dosing regimens may lead to better clinical management of XDR *P. aeruginosa* infections.

**IMPORTANCE** Ceftolozane-tazobactam (C/T) is an antipseudomonal antibiotic with a high clinical impact in treating infection caused by extensively drug-resistant (XDR) *Pseudomonas aeruginosa* isolates, but resistance is emerging. Given its time-dependent behavior, C/T continuous infusion can improve exposure and therefore the pharmacokinetic/pharmacodynamic target attainment. We compared the efficacy of intermittent, extended, and continuous C/T infusion against three XDR ST175 *P. aeruginosa* isolates with different C/T MICs by means of an *in vitro* dynamic hollow-fiber model. We demonstrated that C/T in continuous infusion achieved the largest reduction in bacterial density in the overall treatment arms for both susceptible and resistant isolates. It was also the only regimen with bactericidal activity against all three isolates. Through this study, we want to demonstrate that developing individually tailored antimicrobial treatments is becoming essential. Our results support the role of C/T level monitoring and of dose adjustments for better clinical management and outcomes.

**KEYWORDS** ceftolozane/tazobactam, hollow-fiber, PK/PD, XDR, *Pseudomonas aeruginosa*

Address correspondence to María Milagro Montero, 95422@parcdesalutmar.cat.
The authors declare no conflict of interest.

Antibiotic resistance has led to increased morbidity and mortality worldwide, limiting treatment options and contributing to the emergence and selection of multidrug-resistant (MDR) and extensively drug-resistant (XDR) bacteria (1–3). MDR/XDR *Pseudomonas*

*aeruginosa* isolates are particularly concerning as they are the leading cause of nosocomial infections and are independently associated with in-hospital mortality (4). *P. aeruginosa* has intrinsic resistance to a broad range of antibiotics. This poses a major risk for resistance development due to mutations in chromosomal genes or horizontal gene transfer (1). This is a concern due to the risk of dissemination. Antibiotic resistance is very common in *P. aeruginosa* because of high spontaneous mutation rates, especially in infections with a high bacterial load. *P. aeruginosa* infections pose a medical challenge, particularly when caused by high-risk clones, which are present in hospitals around the world and are directly linked to difficult-to-treat infections (5–7). One example is the ST175 clone, which is particularly common in a number of European countries (8). This clone is also the most common XDR isolate in Spain; the main resistance mechanisms described for ST175 are AmpC hyperproduction and OprD deficiency due to mutations (8). The limited number of treatment options for infections caused by high-risk clones increases the risk of inadequate clinical management. The short-term outlook is not very encouraging due to the lack of a development pipeline for antipseudomonal agents. That said, progress has been made in the development of new molecules in the past year, and new combinations of antibiotics and beta-lactamase inhibitors have appeared.

Ceftolozane-tazobactam (C/T) has emerged as a promising option for treating infections caused by MDR/XDR *P. aeruginosa* isolates resistant to all first-line agents (ticarcillin, piperacillin-tazobactam, ceftazidime, cefepime, aztreonam, imipenem, meropenem, and ciprofloxacin) (9). C/T combines ceftolozane, a novel cephalosporin, with the beta-lactamase inhibitor tazobactam (7). The licensed dosing regimen is a 1-h infusion of 1.5 g every 8 h for complicated urinary tract infections and intra-abdominal infections and 3 g every 8 h for hospital-acquired bacterial pneumonia, including ventilator-associated bacterial pneumonia (10).

Because C/T has time-dependent pharmacokinetic (PK) properties, the most suitable pharmacodynamic (PD) parameter for predicting its bacteriological efficacy is $f\%T > MIC$, which is the percentage of the dosing interval (%T) in which plasma free drug concentrations remain above the MIC. In C/T, this is approximately 40 to 50% (7, 11). The currently recommended dosing regimen might be inadequate to treat infections caused by MDR/XDR *P. aeruginosa* isolates with a C/T MIC above the susceptibility breakpoint of 4 mg/L. In such cases, combination therapy or alternative dosing regimens may need to be individualized to optimize treatment (12).

PK/PD studies are needed to define optimal treatments for XDR *P. aeruginosa* infections. *In vitro* methods can be used to examine interactions between drugs and bacteria to optimize antibiotic use. The hollow-fiber infection model (HFIM) is a dynamic two-compartment method that makes it possible to conduct experiments mimicking human PK under biosafety conditions. It can complement or substitute animal models of infection while overcoming the limitations of static models. In HFIM experiments, bacteria are exposed over time to clinically relevant, fluctuating drug concentrations achieved by repeated dosing and constant elimination.

The aim of this study was to evaluate the efficacy of three C/T dosing regimens against three XDR *P. aeruginosa* isolates in an *in vitro* HFIM. We compared standard intermittent infusion over 1 h, extended infusion over 4 h, and continuous infusion. The isolates were from the ST175 clone and had different C/T MICs (2, 8, and 16 mg/mL).

## RESULTS

***In vitro* susceptibility and resistance mechanisms.** The *P. aeruginosa* (10-023) isolate was susceptible to C/T (MIC, 2 mg/L) and resistant to the other β-lactams due to OprD inactivation and AmpC hyperproduction (13). The *P. aeruginosa* (09-012) isolate was intermediate to C/T (MIC, 8 mg/L), attributable to OprD inactivation, AmpC hyperproduction, and a mutation in PBP3 (R504C) that has been linked to increased β-lactam resistance (13). The *P. aeruginosa* (07-016) isolate was resistant to C/T (MIC, 16 mg/L); in this case, resistance was attributed to production of a class A carbapenemase GES-5 coupled with OprD inactivation (13).

**Estimated frequency of mutants.** The mean density of the *P. aeruginosa* (10-023) (C/T MIC of 2 mg/L) drug-resistant population exposed to C/T at 2, 4, and 8 times the baseline MIC (corresponding to 4, 8, and 16 mg/L) was 1 CFU in $3.3 \times 10^8$, $4.6 \times 10^8$, and $1.6 \times 10^9$

CFU/mL, respectively. The mean density of the *P. aeruginosa* (09-012) (C/T MIC of 8 mg/L) drug-resistant population exposed to C/T at 2, 4, and 8 times the baseline MIC (corresponding to 16, 32, and 64 mg/L) was 1 CFU in $7.2 \times 10^7$, $1.09 \times 10^8$, and $1.15 \times 10^8$ CFU/mL, respectively. Finally, the mean density of the *P. aeruginosa* (07-016) (C/T MIC of 16 mg/L) drug-resistant population exposed to C/T at 2, 4, and 8 times the baseline MIC (corresponding to 32, 64, and 128 mg/L) was 1 CFU in $3.5 \times 10^7$ and $3.75 \times 10^7$ CFU/mL. Mutant frequency could not be determined at concentrations 8 times the baseline MIC due to a lack of growth on the drug-containing plates.

**HFIM.** The mean numbers of bacterial colonies grown for the three C/T regimens over the 7-day HFIM study are shown in Fig. 1. The three isolates, with an initial mean inoculum of 7.26 $\log_{10}$ CFU/mL, were analyzed. Table 1 shows the mean total reduction in the number of colonies (log difference at 24 h) for each regimen compared with that of the control.

The total number of bacterial colonies grown for *P. aeruginosa* (10-023) on day 7 was 4.74 $\pm$ 0.19 $\log_{10}$ CFU/mL for intermittent infusion, 3.53 $\pm$ 0.1 $\log_{10}$ CFU/mL for extended infusion, and 2.54 $\pm$ 0.05 $\log_{10}$ CFU/mL for continuous infusion. The respective reductions were 2.26 $\pm$ 0.19, 3.47 $\pm$ 0.10, and 4.46 $\pm$ 0.05 $\log_{10}$ CFU/mL (Fig. 1A).

The total number of bacterial colonies grown for *P. aeruginosa* (09-012) on day 7 was 4.57 $\pm$ 0.04, 3.91 $\pm$ 0.37, and 1.65 $\pm$ 0.18 $\log_{10}$ CFU/mL for intermittent, extended, and continuous dosing, respectively (Fig. 1B). An overall reduction in number of colonies was observed up to day 6; regrowth was detected in the populations treated with intermittent infusion and extended infusion. The continuous C/T infusion regimen, by contrast, was associated with a continuous reduction. The reductions achieved over the 7 days were 2.53 $\pm$ 0.04 $\log_{10}$ CFU/mL for intermittent infusion, 3.19 $\pm$ 0.37 $\log_{10}$ CFU/mL for extended infusion, and 5.45 $\pm$ 0.18 $\log_{10}$ CFU/mL for continuous infusion.

Finally, the total number of bacterial colonies grown for *P. aeruginosa* (07-016) on day 7 was 6.85 $\pm$ 0.22 $\log_{10}$ CFU/mL for intermittent infusion, 5.98 $\pm$ 0.33 for extended infusion, and 2.74 $\pm$ 0.37 $\log_{10}$ CFU/mL for continuous infusion (Fig. 1C). This C/T-resistant isolate displayed a mean reduction of 4.79 $\log_{10}$ CFU/mL at 8 h for all regimens. There was a regrowth during the first 3 days for the intermittent and extended regimens, and after this point, colony numbers remained relatively stable for the intermittent and extended regimens, with a respective overall reduction of 0.83 $\pm$ 0.22 and 1.7 $\pm$ 0.33 $\log_{10}$ CFU/mL. The corresponding reduction achieved with continuous infusion was 4.94 $\pm$ 0.37 $\log_{10}$ CFU/mL.

Extended infusion had a bactericidal effect on *P. aeruginosa* (10-023) and *P. aeruginosa* (09-012), while intermittent infusion showed no bactericidal activity.

The reductions achieved in overall bacterial density using the different regimens are compared and shown as the log difference on day 7 and the log ratio (LR) of area under the curve for CFU (AUCFU) in Table 1. Relative to control (reference), all regimens achieved greater than 2-log reduction against the three isolates (2.90 to 3.69), and the higher AUCFU reductions were accomplished with the continuous infusion (CI) regimens in the three isolates studied. The regimen of C/T in continuous infusion compared with the regimen of C/T in intermittent infusion achieved greater than 1-log reduction against the three isolates. Compared with the regimen of C/T in extended infusion, the reductions accomplished with C/T in continuous infusion were greater than 1-log against *P. aeruginosa* (09-012) and *P. aeruginosa* (07-016). Referenced to the reduction achieved with C/T in extended infusion versus intermittent infusion, the reductions were less than 1-log reduction (0.10 to 0.87), against the three isolates. Regarding the $f\%T > MIC$ parameter, in the overall regimens of the HFIM performed, the $f\%T > MIC$ accomplished was greater than 98%.

**Drug concentrations.** The relationship between observed and predicted C/T concentrations over the 7 days is shown in Table 2. The simulated drug exposures were satisfactory, with $R^2$ values of 0.920 for intermittent infusion, 0.921 for extended infusion, and 0.905 for continuous infusion (Fig. 2). For each point of predicted C/T concentration, the observed concentration increased by between 1.03 and 1.05 (mg/L).

## DISCUSSION

Optimization of antimicrobial PK/PD properties when treating XDR *P. aeruginosa* infections is crucial in our setting. The appearance of novel antibiotic products such as

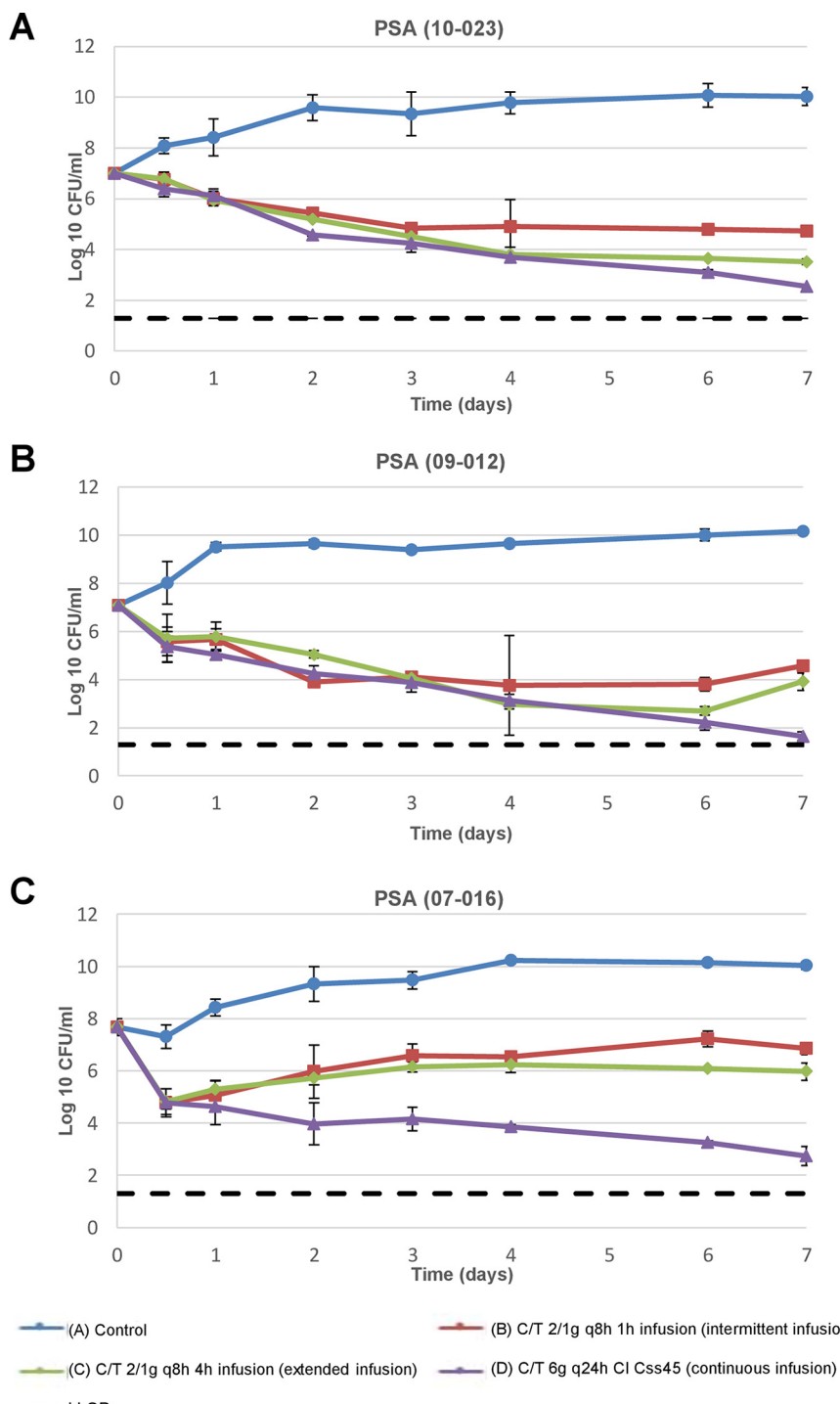

**FIG 1** Mean reduction in number of bacterial colonies (CFU/mL) over 7 days in an *in vitro* HFIM model testing three C/T infusion regimens (1-h, 4-h, and continuous infusion) against three XDR ST175 *P. aeruginosa* isolates: (A) *P. aeruginosa* (10-023), (B) *P. aeruginosa* (09-012), and (C) *P. aeruginosa* (07-016), with respective C/T MIC values of 2, 8, and 16 mg/L. Experiments were conducted in duplicate. C/T, ceftolozane/tazobactam; CI, continuous infusion; Css, steady-state concentration; q8h, every 8 h; LLOD, lower limit of detection.

C/T paves the way toward a treatment era in which individual characteristics will be taken into account to achieve optimized strategies. C/T is a promising option for the treatment of *P. aeruginosa* infections (14). Because it is a time-dependent antimicrobial, extended infusion could improve the probability of optimal PK/PD target attainment.

**TABLE 1** Mean overall reduction in number of bacterial colonies grown with alternative C/T infusion regimens for each ST175 isolate; parameters included $\log_{10}$ CFU/mL $\pm$ standard deviation, and LR of AUCFU

| Infusion regimen | *P. aeruginosa* (10-023) | | *P. aeruginosa* (09-012) | | *P. aeruginosa* (07-016) | |
|---|---|---|---|---|---|---|
| | Log diff day 7[a] | LR of AUCFU[b] | Log diff day 7 | LR of AUCFU | Log diff day 7 | LR of AUCFU |
| C/T 2/1 g q8h 1-h infusion vs control | −2.26 ± 0.19 | −3.37 | −2.53 ± 0.04 | −3.66 | −0.83 ± 0.22 | −2.90 |
| C/T 2/1 g q8h 4-h infusion vs control | −3.47 ± 0.10 | −3.38 | −3.19 ± 0.37 | −3.64 | −1.7 ± 0.33 | −3.15 |
| C/T 6 g q24h CI Css45 vs control | −4.46 ± 0.05 | −3.53 | −5.45 ± 0.18 | 3.69 | −4.94 ± 0.37 | −3.24 |
| C/T 6 g q24h CI Css45 vs C/T 2/1 g q8h 1-h infusion | −2.2 ± 0.1 | −1.01 | −2.92 ± 0.01 | −1.52 | −4.11 ± 0.12 | −2.1 |
| C/T 6 g q24h CI Css45 vs C/T 2/1 g q8h 4-h infusion | −0.99 ± 0.33 | −0.65 | −2.26 ± 0.2 | −1.23 | −3.24 ± 0.05 | −1.85 |
| C/T 2/1 g q8h 4-h infusion vs C/T 2/1 g q8h 1-h infusion | −1.21 ± 0.05 | −0.87 | −0.66 ± 0.15 | −0.10 | −0.87 ± 0.28 | −0.15 |

[a]Log difference at the end of the assay for each regimen compared with the control.
[b]The log difference is presented as the log ratio (LR), which is used to compare any number of $\log_{10}$ CFU of two regimens (test/reference). AUCFU, area under the curve for CFU; C/T, ceftolozane-tazobactam; CI, continuous infusion; Css, steady-state concentration; q8h, every 8 h.

We used the HFIM system to compare three C/T infusion regimens against three XDR *P. aeruginosa* ST175 isolates with C/T MIC values ranging from 2 to 16 mg/L. The ST175 clone has been associated with MDR/XDR isolates; it is a common hospital contaminant and causes difficult-to-treat respiratory tract infections in patients with cystic fibrosis or chronic obstructive pulmonary disease (14, 15).

Our *in vitro* study showed that overall, C/T in continuous infusion reduced the density of susceptible and resistant *P. aeruginosa* isolates, reinforcing the idea that this mode of administration results in concentrations that remain above the susceptibility breakpoint for longer. The reduction in density was even more evident in less-susceptible isolates with higher MIC values, where continuous infusion of C/T led to sustained suppression of the bacterial population, outperforming both the intermittent and extended dosing regimens. In addition, it was the only regimen with bactericidal activity against all three isolates, supporting its potential superiority and suggesting that the currently recommended regimen does not provide adequate coverage against MDR/XDR *P. aeruginosa* isolates. Our results are consistent with those reported by Pilmis et al. (16), who found that compared with intermittent administration, C/T in continuous infusion was associated with a higher probability of target attainment (>90%) for MDR *P. aeruginosa* isolates with a C/T MIC of 4 mg/L. Sime et al. (10) showed that a C/T dosing regimen of 3 g every 8 h was associated with relatively low fractional target attainment in patients with severe augmented renal clearance, which could be problematic when treating *P. aeruginosa* infections caused by MDR isolates that are potentially less susceptible to this antibiotic combination (17, 18).

Findings from other PK/PD simulation studies suggest that optimal $\beta$-lactam exposure is rapidly obtained via continuous or extended infusion (19, 20). Natesan et al. (12), using Monte Carlo simulation to determine which C/T dosing regimens were most likely to optimize probability of target attainment for MDR *P. aeruginosa* isolates with different C/T MICs, found that extended infusion was superior in certain scenarios. In our study, the extended dosing regimen showed only a slight advantage over the currently recommended 1-h regimen (final mean number of bacterial colonies of 5.39 versus 4.48 CFU/mL).

If it is confirmed that continuous infusion of C/T achieves the greatest reduction in MDR/XRD *P. aeruginosa* populations, optimization of steady-state concentrations will be necessary to achieve optimal clinical outcomes and prevent the selection of C/T-resistant subpopulations (21, 22). In a previous study by our group, a steady-state concentration of 45 mg/L reduced bacterial density and prevented the emergence of C/T resistance in HFIM assays

**TABLE 2** Observed versus predicted antibiotic concentrations achieved in each HFIM model[a]

| Dosing regimen | Free peak concn (mg/L) ± SD | | Free trough concn (mg/L)/Css ± SD | |
|---|---|---|---|---|
| | Predicted value | Observed value | Predicted value | Observed value |
| C/T 2/1 g q8h 1-h infusion | 74.45 | 61.96 ± 6.80 | 14.77 | 25.67 ± 3.7 |
| C/T 2/1 g q8h 4-h infusion | 54.55 | 53.10 ± 7.92 | 19.7 | 27.29 ± 5.63 |
| C/T 6 g q24h CI | | | 45 | 47.29 ± 5.43 |

[a]Data are presented as the mean concentration ± standard deviation. Css, steady-state concentration; q8h 1-h, infusion over 1 h every 8 h (intermittent infusion); q8h 4-h, infusion over 4 h every 8 h (extended infusion); SD, standard deviation.

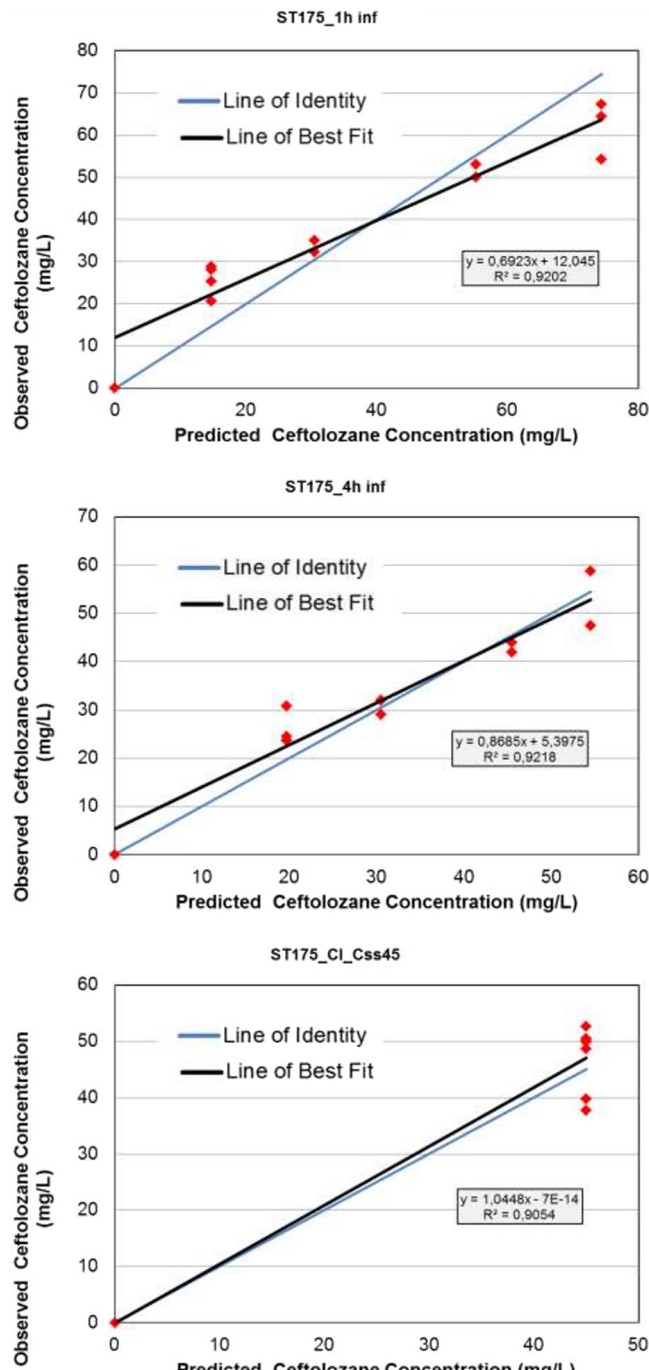

**FIG 2** Relationship between observed and predicted ceftolozane concentrations for intermittent (1-h), extended (4-h), and continuous C/T infusion regimens in the overall experiments. inf, infusion; CI, continuous infusion; Css, steady-state concentration.

with XDR *P. aeruginosa* isolates (22). Escolà-Vergé et al. (23) reported the development of resistance with the use of low-dose (1.5 g every 8 h) and high-dose (3 g every 8 h) C/T in the treatment of MDR *P. aeruginosa* infections. The increase in MIC values ranged from 8-fold to >85-fold.

Although the study parameters (dose, dosing interval, and duration of experiment) were designed to simulate clinical exposure, our study has a number of limitations that should be taken into account. Because it was an *in vitro* study, we were unable to examine toxicity, immune responses, or injection-site PK/PD effects. Moreover, some PK parameters,

such as half-life and clearance, could be altered in some critically ill patients. We also studied just three isolates, although they are representative of C/T susceptibility ranges in our environment.

In summary, our findings show that C/T in continuous infusion achieves a greater overall reduction in bacterial burden than intermittent or extended dosing regimens, particularly in the case of nonsusceptible XDR *P. aeruginosa* isolates. The current recommended dosing regimen would appear to offer inadequate coverage for optimal PK/PD target attainment. Continuous infusion regimens are potentially useful and should be investigated further. The findings of this *in vitro* study suggest that correct use of C/T dosing regimens could lead to better clinical management of *P. aeruginosa* infections caused by XDR isolates.

## MATERIALS AND METHODS

**Bacterial isolates and resistance mechanisms.** We selected three XDR ST175 *P. aeruginosa* clinical isolates representative of the clones and resistance mechanisms in our environment: *P. aeruginosa* (10-023), *P. aeruginosa* (09-012), and *P. aeruginosa* (07-016). The isolates were collected from a collection of 150 XDR clinical isolates from nine hospitals located in six different Spanish regions in the context of a multicenter clinical study (EudraCT 2013-005583-25, PI JP Horcajada). They were obtained from different infection sources and stored at −80°C (storage vials with 10% glycerol). Fresh isolates were subcultured twice on 5% blood agar plates for 24 h at 35°C before each experiment. The isolates had been previously characterized for molecular epidemiology purposes using pulsed-field gel electrophoresis (PFGE), multilocus sequence typing (MLST), and whole-genome sequencing (13). Expression levels of *ampC*, *oprD*, *mexB*, *mexD*, *mexF*, and/or *mexY* were determined using RT-PCR (reverse transcription PCR); expression of outer membrane proteins (including OprD) was determined by SDS-PAGE, while penicillin-binding protein (PBP) profiles were determined in a competition assay with fluorescent penicillin (Bocillin Fl). The main target mutations were sequenced using previously described primers. Clonal relatedness was evaluated by PFGE. Finally, whole-genome sequencing was performed in the three isolates (13).

**Antibiotics.** C/T (Zerbaxa; Merck & Co., Inc., Kenilworth, NJ; lot number SO15404; expiration date, August 2020) was provided by Merck & Co., Inc. (Kenilworth, NJ). The antibiotic solutions were prepared according to CLSI guidelines (24). The dosing regimens were within a dose range based on previously determined maximum concentration of drug in serum ($C_{max}$) and area under the curve (AUC) (21). Three regimens were simulated: 2 g/1 g every 8 h over 1 h (intermittent infusion) to reach a free $C_{max}$ ($fC_{max}$) of 75 mg/L, 2 g/1 g every 8 h over 4 h (extended infusion) to reach an $fC_{max}$ of 55 mg/L (10), and continuous infusion to reach a steady-state concentration of 45 mg/L. The simulated elimination half-life for ceftolozane was 3 h (14, 21). Although tazobactam is present in the pharmaceutical formulation, exposure to this drug was not considered since it has a limited role in ceftolozane's activity against *P. aeruginosa* (25). C/T concentrations were validated by high-performance liquid chromatography (HPLC) (26).

**Estimated frequency of spontaneous mutants.** Preexisting mutations in genes associated with C/T resistance may increase the risk of C/T-resistant subpopulations developing in the total bacterial population. The frequency of spontaneous mutants conferring C/T resistance was estimated for all isolates by plating 4 mL of log-phase growth suspension onto agar containing ceftolozane at concentrations of 2, 4, and 8 times the baseline MIC and tazobactam at a fixed concentration of 4 mg/L. The experiments were performed in duplicate. After 48 h of incubation, the bacterial concentration within each suspension was determined by quantitative culture. The ratio of growth on the drug-containing plates to that of the starting inoculum provided an estimate of the frequency of mutants conferring drug resistance within each population.

***In vitro* antimicrobial susceptibility testing.** Antimicrobial susceptibility testing was performed according to the CLSI guidelines (24) for broth microdilution using cation-adjusted Mueller-Hinton broth (CAMHB). The ceftolozane susceptibility test was conducted alone and in combination with a fixed tazobactam concentration (4 mg/L).

**HFIM.** The efficacy of the three C/T infusion regimens against *P. aeruginosa* (10-023), *P. aeruginosa* (09-012), and *P. aeruginosa* (07-016) was investigated in a 7-day HFIM study as described previously (14, 27). In the HFIM, bacteria are exposed over time to clinically relevant, fluctuating drug concentrations achieved by repeated dosing and constant elimination. Four arms were analyzed: no treatment (control), intermittent infusion, extended infusion, and continuous infusion. Polyethersulfone hemofilters were used as the hollow-fiber cartridges with a volume of 50 mL (Aquamax HF03, Nikkiso, Belgium). Experiments were conducted in duplicate at 37°C in a humidified incubator set. Separate infusion pumps were used to pump each of the C/T regimens into the central reservoir to reach predicted concentrations simulating free drug PK profiles in humans. Fresh drug-free growth medium CAMHB was continuously infused into the central reservoir to dilute and simulate drug elimination in humans. An equal volume of drug-containing medium was concurrently removed from the central reservoir to maintain an isovolumetric system. An overnight culture of each isolate was diluted with CAMHB and further incubated at 37°C in a water bath shaker to reach early log-phase growth. The density of the growth broth was calculated for an initial inoculum of $10^7$ to $10^8$ CFU/mL using a spectrophotometer at 630 nm. The extracapillary space of each HFIM was inoculated with 50 mL of the bacterial suspension to simulate high-inoculum infections. The bacteria were confined to the extracapillary space but exposed to fluctuating drug concentrations from the HFIM cartridge through an internal circulatory pump in the bioreactor loop. At 0, 8, 24, 48, 72, 96, 144, and 168 h, bacterial samples were collected from the cartridges, washed, centrifuged twice at 13,000 rpm for 3 min, and then reconstituted with sterile saline solution to the same original volume

to minimize drug carryover. Serially diluted samples were quantitatively cultured onto drug-free Trypticase soy agar (BBL TSA II, Becton, Dickinson) plates to determine the count of the total bacterial population ($\log_{10}$ CFU/mL). The inoculated plates were incubated in a humidified incubator (37°C) for 24 h, the bacterial colonies were visually counted, and the number of bacterial cells in the original sample was calculated based on the dilution factor. The lower limit of detection (LLOD) was 1.3 $\log_{10}$ CFU/mL. Bactericidal activity was defined as a reduction of 3 $\log_{10}$ CFU/mL from the initial bacterial density (28).

**Drug concentrations.** Antibiotic samples were collected from the peripheral compartment of the HFIM system at different time points over the first 48 h and once a day until the end of the study. Samples were stored at −80°C until analysis. Samples were taken to validate ceftolozane concentrations. All exposures simulating free drug PK in humans were based on the half-life of ceftolozane. Samples were taken to report free peak and trough antibiotic values, and concentrations were analyzed by HPLC (26).

**Descriptive statistics.** The difference in area under the curve for CFU (AUCFU) was calculated as described previously (29). We calculated an end-of-treatment endpoint, the log ratio of AUCFU (LR), as LR = $\log_{10}$ (AUCFUtest/AUCFUreference), where the reference regimen was the growth control. An LR value of −1 indicated a 90% (10-fold) reduction in overall bacterial density, while a value of −2 indicated a 99% (100-fold) reduction (28, 29).

## ACKNOWLEDGMENTS

We thank The Institute for Clinical Pharmacodynamics (ICPD), Schenectady, NY and the Infectious Pathology and Antimicrobials Research Group (IPAR), Institute Hospital del Mar d'Investigacions Mèdiques (IMIM) for their support.

We declare no conflict of interest.

Conceptualization: Maria Milagro Montero; methodology: Maria Milagro Montero, Sandra Domene, Núria Prim, Lorena Ferrer Alapont; formal analysis and investigation: Maria Milagro Montero, Sandra Domene; writing—original draft preparation: Maria Milagro Montero, Sandra Domene, Lorena Ferrer Alapont; review and editing: all authors; funding acquisition: Maria Milagro Montero, Juan Pablo Horcajada; supervision: all authors. All authors have read and agreed to the published version of the manuscript.

This study was partially supported by the Ministerio de Economía y Competitividad of Spain, Instituto de Salud Carlos III FEDER PI16/00669, PI17/00251, PI18/0076, and the Spanish Network for Research in Infectious Diseases (REIPI RD16/0016).

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
