## [Reviewer comments · Microbiology Spectrum]

Microbiology Spectrum

Comparison of ceftolozane/tazobactam infusion regimens in a hollow-fiber infection model against extensively drug-resistant *Pseudomonas aeruginosa* isolates

Maria Montero, Sandra Domene Ochoa, Carla López-Causapé, Inmaculada López Montesinos, Sonia Luque, María Luisa Sorlí, Nuria Campillo, Eduardo Padilla, Nuria Prim, Lorena Ferrer Alapont, Santiago Grau-Cerrato, Antonio Oliver, and Juan P. Horcajada

Corresponding Author(s): Maria Montero, Hospital del Mar. Institut Hospital del Mar d'Investigacions Mèdiques

Review Timeline:

Submission Date:	March 9, 2022
Editorial Decision:	April 11, 2022
Revision Received:	May 19, 2022
Accepted:	May 20, 2022

Editor: Aude Ferran

Reviewer(s): Disclosure of reviewer identity is with reference to reviewer comments included in decision letter(s). The following individuals involved in review of your submission have agreed to reveal their identity: Gloria Wong (Reviewer #2)

Transaction Report:

DOI: <https://doi.org/10.1128/spectrum.00892-22>

April 11, 2022

Prof. Maria Milagro Montero
Hospital del Mar. Institut Hospital del Mar d'Investigacions Mèdiques
Infectious diseases
paseo maritimo 25-29
barcelona 08003
Spain

Re: Spectrum00892-22 (Comparison of ceftolozane/tazobactam infusion regimens in a hollow-fiber infection model against extensively drug-resistant *Pseudomonas aeruginosa* isolates)

Dear Prof. Maria Milagro Montero:

Thank you for submitting your manuscript to Microbiology Spectrum. Two reviewers considered that your manuscript needs revision (you will find the comments at the end of the message). The reviewer 1's comments are particularly relevant and should be addressed.

Link Not Available

Sincerely,

Aude Ferran

Journals Department
Reviewer comments:

Reviewer #1 (Comments for the Author):

The manuscript presents hollow-fiber infection model (HFIM) experiments exposing three *P. aeruginosa* (ST175) isolates with previously characterized resistance mechanisms to three different ceftolozane/tazobactam (C/T) dosing regimens for 7 days: a 1

h intermittent i.v. infusion every 8 h (2 g cef/1 g taz), a 4 h extended i.v. infusion (2 g cef/1 g taz), and a continuous infusion to reach a steady-state ceftozolane concentration of 45 mg/L.

As the largest extent of bacterial killing was observed under exposure to the continuous infusion for the three isolates, authors conclude that this regimen would be superior to the licensed intermittent infusion regimens and could improve clinical outcome even for resistant strains.

The manuscript addresses a relevant question and presents interesting data. However, even though antibiotic concentrations in the HFIM experiments were determined, C/T concentrations over time are not reported, impeding profound pharmacokinetic/pharmacodynamics (PK/PD) analysis. Data analysis and discussion remain rather shallow. Particularly, linking the "real", i.e. measured exposure in the experiments to the bacterial growth and kill behavior could enable a more comprehensive characterization of the PK/PD relationship and comparison between the strains in the light of their resistance mechanisms.

Specific comments

Introduction

L.89: %T>MIC is a PK/PD parameter (not a "PD parameter"); it relates an exposure metric to the MIC, please correct and use "f%T>MIC" to refer to free (i.e. unbound) plasma concentrations

Material and methods

LL 223/224: It is unclear how the targeted C_{max} and AUC values were derived and how they are linked to the mimicked dosing regimens. If a PK model was applied, please shortly characterize the model and the underlying patient population; or elaborate on any other applied approach to answer the question whether the mimicked drug concentration-time (C(t)) profiles are clinically relevant.

LL 228/229: To get the full picture, reporting both ceftolozane and tazobactam C(t) profiles mimicked in the HFIM would still be of interest.

LL 234 - 236: The MIC values beyond C/T are not reported and not relevant, can be omitted.

L 249 (referring to LL 223/224): How was mimicking of free C(t) profiles assured (based on which kind of clinical data, i.e. measured free concentrations or assuming a certain protein binding?)

L 254: typo: CFU/mL (not "UFC/mL")

L 271 - 273: Please clarify if only cef or both cef and taz concentrations were determined and report both over time, if possible.

Results

LL 109-114: Please clarify whether the reported MIC values refer to ceftolozane with or without tazobactam addition (LL 238/239: "The ceftolozane susceptibility test was conducted alone and in combination with a fixed tazobactam concentration"). Further, the breakpoint value and the applied classification should be reported (EUCAST/CLSI?) and the category of PSA(09-012) should be clarified: intermediate, resistant, or "susceptibly, increased exposure" (in the EUCAST classification)?

LL 120-143: Bacterial concentrations at day 7 are described quite extensively, while these numbers can be extracted from Figure 1 easily and represent only one specific timepoint. As the AUCFU ratios were also determined, it would be more interesting to focus on these values, as they represent the overall effect over 7 days. Moreover, it would be interesting to link the observed overall effect (AUCFU) to the respective PK/PD parameters, such as f %T>MIC (based on measured C/T concentrations), to learn more about the PK/PD relationship and the differences between the three dosing regimens and strains.

L 127: There is only one value (or a very small error bar?) for PSA(09-12) under exposure to an intermittent infusion at day 7, which makes it very uncertain to interpret "regrowth", as the 4 previous values don't show any trend toward killing or regrowth. At least one more replicate is needed to confirm this observation.

LL 134/135: I disagree that colony numbers "remained relatively stable", as Figure 1C shows regrowth under exposure to the extended and the intermittent infusion during the first 3 days.

LL 139/140: I don't think one can conclude that intermittent infusion showed no activity at all, as Figure 1A and 1B show a reduction of bacterial concentrations during the first days for all dosing regimens.

LL 145-149: It would be more informative to show the targeted and the measured concentrations over time for each replicate, to enable an evaluation of the potential impact of PK variability on the results. This might especially be relevant for the intermittent infusion, as there are some relevant deviations between the observed and the predicted concentrations. For the continuous infusion, the linear regression is not meaningful. It would be more interesting to see whether a trend over time is observable.

Discussion

The discussed conclusions are overall plausible, but some relevant points are missing:

- For PSA(09-12) and PSA (07-16), (potential) regrowth was observed. Based on the reported resistance mechanisms, are there any mechanistic insights into the drivers of regrowth and the link to the respective PK, also in the light of the determined PK parameters, such as f %T>MIC?
- Based on an extended PK/PD analysis, does the data support a certain f %T>MIC target value?
- L 194: What is meant by "injection-site PK/PD effects"? Do you refer to the limitation that plasma is typically not the target-site and it would be more informative to mimic target-site PK in HFIM experiments?
- From the presented data, the superiority of continuous infusions seems obvious. Why is it not recommended/licensed/used in the clinics - are there toxicity concerns? Why?

Figure 1:

To follow the description of the time-kill curves more easily, it would be helpful for the reader to add the terms "intermittent infusion", "extended infusions" and "continuous infusion" to the legend, as these are used in the text. Further, the numbers of replicates summarized in the mean values should be added in the caption.

Reviewer #2 (Comments for the Author):

Thank you for the opportunity to review this manuscript. I think it's a neat small study. Although not novel, the results presented add to the body of literature around importance of dosing regimen and PK/PD considerations especially dealing with resistance microbes.

A few comments for the authors to consider:

1. I am interested to see the data in table 1 to include LR of AUCFU between continuous infusion vs intermittent regimens, ie q24hr vs 1hr infusion and q24hr vs 4hr infusion, rather than just versus control. I think that can conceptualise the magnitude of differences between the regimens better, as depicted in Figure 1.
2. Figure 2 - although the "overall" predicted concentrations agrees well with observed concentrations, I noted in Table 2, the data demonstrated a slightly difference picture. The observed concentrations are higher than predicted for trough concentrations for intermittent dosing (almost double in 1hr infusion), which is likely secondary to poorer predictive values at lower end of concentrations (hence the poorer fit in figure 2 first graph). I think the data for peak and trough concentrations in figure 2 should presented separately as in Table 2 to be consistent. The implication here in beta-lactate is unlikely huge given the large therapeutic window, however the issues with under-predictions with modelling should not be ignored
3. Page 7 paragraph from line 192 - another limitation is the assumptions of PK parameters such as half life and clearance used, which is likely altered in some patients especially the critically ill

Staff Comments:

Preparing Revision Guidelines

Please return the manuscript within 60 days; if you cannot complete the modification within this time period, please contact me. If you do not wish to modify the manuscript and prefer to submit it to another journal, please notify me of your decision immediately so that the manuscript may be formally withdrawn from consideration by Microbiology Spectrum.

Comparison of ceftolozane/tazobactam infusion regimens in a hollow-fiber infection model against extensively drug-resistant *Pseudomonas aeruginosa* isolates

Reviewer #1 (Comments for the Author):

The manuscript presents hollow-fiber infection model (HFIM) experiments exposing three *P. aeruginosa* (ST175) isolates with previously characterized resistance mechanisms to three different ceftolozane/tazobactam (C/T) dosing regimens for 7 days: a 1 h intermittent i.v. infusion every 8 h (2 g cef/1 g taz), a 4 h extended i.v. infusion (2 g cef/1 g taz), and a continuous infusion to reach a steady-state ceftozolane concentration of 45 mg/L.

As the largest extent of bacterial killing was observed under exposure to the continuous infusion for the three isolates, authors conclude that this regimen would be superior to the licensed intermittent infusion regimens and could improve clinical outcome even for resistant strains.

The manuscript addresses a relevant question and presents interesting data. However, even though antibiotic concentrations in the HFIM experiments were determined, C/T concentrations over time are not reported, impeding profound pharmacokinetic/pharmacodynamics (PK/PD) analysis. Data analysis and discussion remain rather shallow. Particularly, linking the "real", i.e. measured exposure in the experiments to the bacterial growth and kill behavior could enable a more comprehensive characterization of the PK/PD relationship and comparison between the strains in the light of their resistance mechanisms.

Introduction

L.89: %T>MIC is a PK/PD parameter (not a "PD parameter"); it relates an exposure metric to the MIC, please correct and use "f%T>MIC" to refer to free (i.e. unbound) plasma concentrations

- Done

Material and methods

LL 223/224: It is unclear how the targeted Cmax and AUC values were derived and how they are linked to the mimicked dosing regimens. If a PK model was

applied, please shortly characterize the model and the underlying patient population; or elaborate on any other applied approach to answer the question whether the mimicked drug concentration-time ($C(t)$) profiles are clinically relevant.

- The targeted C_{max} and AUC were obtained from the cited bibliography.

LL 228/229: To get the full picture, reporting both ceftolozane and tazobactam $C(t)$ profiles mimicked in the HFIM would still be of interest.

- Although tazobactam is present in the pharmaceutical formulation, exposure to this drug was not considered since it has a limited role in ceftolozane's activity against *P. aeruginosa* [25] [29]. (line 259)

Refs:

- Zhanel GG, Chung P, Adam H, Zelenitsky S, Denisuik A, Schweizer F, et al. Ceftolozane/Tazobactam: A Novel Cephalosporin/ β -Lactamase Inhibitor Combination with Activity Against Multidrug-Resistant Gram-Negative Bacilli. *Drugs* 2014;74:31–51. doi: 10.1007/s40265-013-0168-2
- Rico Caballero V, Almarzoky Abuhussain S, Kuti JL, Nicolau DP. Efficacy of Human-Simulated Exposures of Ceftolozane-Tazobactam Alone and in Combination with Amikacin or Colistin against Multidrug-Resistant *Pseudomonas aeruginosa* in an In Vitro Pharmacodynamic Model. *Antimicrob. Agents Chemother.* 2018;62. doi: 10.1128/AAC.02384-17

LL 234 - 236: The MIC values beyond C/T are not reported and not relevant, can be omitted.

- It has been removed from the text

L 249 (referring to LL 223/224): How was mimicking of free $C(t)$ profiles assured (based on which kind of clinical data, i.e. measured free concentrations or assuming a certain protein binding?)

- The simulated free C/T profiles were measured based on certain protein binding (assuming an 40% of protein binding in the case of ceftolozane)

L 254: typo: CFU/mL (not "UFC/mL")

- Done

L 271 - 273: Please clarify if only cef or both cef and taz concentrations were determined and report both over time, if possible.

- The concentrations determined were of ceftolozane (due to, as mentioned previously, the limited role of tazobactam on ceftolozane's activity against *P. aeruginosa*)

The mean concentrations of ceftolozane achieved in each HFIM over time were presented in the Table 2. We presented data about the C_{max} (free peak concentration) and C_{min} (free trough concentration), and the steady-state concentration for the continuous infusion.

For your interest, there were the different values of C_{max}, C_{min} and C_{ss} achieved over time:

- For the 1h infusion:
 - C_{max}: 55.8, 86.1, 51.3, 50.8, 62.5, 49.5, 66.8, 75.6, 59.2
 - C_{min}: 30.6, 17.3, 13.9, 34, 22.7, 29.5, 21, 26.6, 28.3, 26.5, 29.6
- For the 4h infusion:
 - C_{max}: 41.6, 48.7, 52.2, 81.1, 44.9, 50.1
 - C_{min}: 24, 24.7, 24.6, 30.5, 28.5, 29.4
- For the CI:
 - C_{ss}: 34.4, 36.8, 43.2, 40.5, 45, 39.4, 44.2, 42.7, 42.2, 42.4, 58.5, 53.8, 54, 54.5, 54.9, 54.7, 36.65, 40.05, 48.4, 63.7, 52.35, 51.85, 50.55, 50.15

Results

LL 109-114: Please clarify whether the reported MIC values refer to ceftolozane with or without tazobactam addition (LL 238/239: "The ceftolozane susceptibility test was conducted alone and in combination with a fixed tazobactam concentration"). Further, the breakpoint value and the applied classification should be reported (EUCAST/CLSI?) and the category of PSA (09-012) should be clarified: intermediate, resistant, or "susceptibly, increased exposure" (in the EUCAST classification)?

- The reported MIC values referred to ceftolozane with tazobactam addition. The applied classification was according to CLSI guidelines (line 256 of the methods section). The PSA (09-012) was classified as intermediate to C/T, according to CLSI guidelines.

LL 120-143: Bacterial concentrations at day 7 are described quite extensively, while these numbers can be extracted from Figure 1 easily and represent only one specific timepoint. As the AUCFU ratios were also determined, it would be more interesting to focus on these values, as they represent the overall effect over 7 days. Moreover, it would be interesting to link the observed overall effect (AUCFU) to the respective PK/PD parameters, such as $f\%T>MIC$ (based on measured C/T concentrations), to learn more about the PK/PD relationship and the differences between the three dosing regimens and strains.

- Done, we added some information regarding AUCFU in the results section. We also added AUCFU results comparing the different C/T regimens.

L 127: There is only one value (or a very small error bar?) for PSA (09-12) under exposure to an intermittent infusion at day 7, which makes it very uncertain to interpret "regrowth", as the 4 previous values don't show any trend toward killing or regrowth. At least one more replicate is needed to confirm this observation.

- A replicate was done. There were two values, but it had a very small error bar. A reduction in number of colonies was observed up to day 6, and from that point onwards a small regrowth was observed (from 3.80 at day 6 to 4.57 at day 7)

LL 134/135: I disagree that colony numbers "remained relatively stable", as Figure 1C shows regrowth under exposure to the extended and the intermittent infusion during the first 3 days.

- It was corrected.

LL 139/140: I don't think one can conclude that intermittent infusion showed no

activity at all, as Figure 1A and 1B show a reduction of bacterial concentrations during the first days for all dosing regimens.

- It was corrected. We were referring to bactericidal activity (defined as a reduction of 3 log₁₀ CFU/mL from the initial bacterial density).

LL 145-149: It would be more informative to show the targeted and the measured concentrations over time for each replicate, to enable an evaluation of the potential impact of PK variability on the results. This might especially be relevant for the intermittent infusion, as there are some relevant deviations between the observed and the predicted concentrations. For the continuous infusion, the linear regression is not meaningful. It would be more interesting to see whether a trend over time is observable.

- The mean concentration was informed on the table 2 for all regimens (with the corresponding standard deviation). We collected a large number of concentrations over time, for this reason we considered that the mean could be more clearly. Anyway, we added previously, for your interest, the achieved C_{max}, C_{min} and C_{ss} (on a previous comment).
- We agree with the comment of the continuous infusion graph. Anyway, we think that giving the same view for all three types of graphs is interesting, and maybe the trend over time representation would be a part of the supplemental material. We represented the linear regression for your interest.

Discussion

The discussed conclusions are overall plausible, but some relevant points are missing:

For PSA (09-12) and PSA (07-16), (potential) regrowth was observed. Based on the reported resistance mechanisms, are there any mechanistic insights into the drivers of regrowth and the link to the respective PK, also in the light of the determined PK parameters, such as $f\%T>MIC$?

- In this study, it was not confirmed what had happened with the regrowth. In previous works of our group, the development of resistance was studied according to the PK concentrations of C/T (due to mutations in AmpC). At lower concentrations, resistance mutations arose. Reference: Montero MM, Domene-Ochoa S, López-Causapé C, Luque S, Sorlí L, Campillo N, et al. Impact of ceftolozane/tazobactam concentrations in continuous infusion against extensively drug-resistant *Pseudomonas aeruginosa* isolates in a hollow-fiber infection model. *Sci. Reports* 2021 11:12;11:1–8. doi: 10.1038/s41598-021-01784-4

Based on an extended PK/PD analysis, does the data support a certain $f\%T>MIC$ target value?

- Yes. Average percentages of time above the MIC ($f\%T>MIC$) were calculated and are displayed in the table. In the overall regimens of the HFIM performed the $f\%T>MIC$ accomplished was greater than 98%.

	Average $f\%T>MIC$		
	1h	4h	CI
(10_023)	98.6	99.45	99.99
(09_012)	98.96	99.79	99.17
(07_016)	99.27	99.48	98.16

L 194: What is meant by "injection-site PK/PD effects"? Do you refer to the limitation that plasma is typically not the target-site and it would be more informative to mimic target-site PK in HFIM experiments?

- Yes, we referred to the PK/PD effects occurring at the specific site of an infection.

From the presented data, the superiority of continuous infusions seems obvious. Why is it not recommended/licensed/used in the clinics - are there toxicity concerns? Why?

- Continuous infusion of C/T is used in the clinics. Currently, given that there is increasing scientific evidence for the use of C/T in continuous infusion, there is a tendency to use C/T in continuous infusion in most cases of serious infections.

Figure 1:

To follow the description of the time-kill curves more easily, it would be helpful for the reader to add the terms "intermittent infusion", "extended infusions" and "continuous infusion" to the legend, as these are used in the text. Further, the numbers of replicates summarized in the mean values should be added in the caption.

- Done

Reviewer #2 (Comments for the Author):

Thank you for the opportunity to review this manuscript. I think it's a neat small study. Although not novel, the results presented add to the body of literature around importance of dosing regimen and PK/PD considerations especially dealing with resistance microbes.

A few comments for the authors to consider:

1. I am interested to see the data in table 1 to include LR of AUCFU between continuous infusion vs intermittent regimens, ie q24hr vs 1hr infusion and q24hr vs 4hr infusion, rather than just versus control. I think that can conceptualise the magnitude of differences between the regimens better, as depicted in Figure 1.

- Done

2. Figure 2 - although the "overall" predicted concentrations agrees well with observed concentrations, I noted in Table 2, the data demonstrated a slightly difference picture. The observed concentrations are higher than predicted for trough concentrations for intermittent dosing (almost double in 1hr infusion),

which is likely secondary to poorer predictive values at lower end of concentrations (hence the poorer fit in figure 2 first graph). I think the data for peak and trough concentrations in figure 2 should be presented separately as in Table 2 to be consistent. The implication here in beta-lactate is unlikely huge given the large therapeutic window, however the issues with under-predictions with modelling should not be ignored

- Figure 2 was made to give an overview of the different PK results throughout the experiment, not just the trough and peak concentrations. For this reason, it has been represented together in a graph. The table sought to give the vision of the values separately. We believe that it was a way to give as much information as possible, and it is the model used in previous articles, for example:

- Montero M, VanScoy BD, López-Causapé C, Conde H, Adams J, Segura C, et al. Evaluation of ceftolozane-tazobactam in combination with meropenem against *Pseudomonas aeruginosa* sequence type 175 in a hollow-fiber infection model. *Antimicrob. Agents Chemother.* 2018;62. doi: 10.1128/AAC.00026-18

- Montero M, Ochoa SD, López-Causapé C, VanScoy B, Luque S, Sorlí L, et al. Efficacy of ceftolozane-tazobactam in combination with colistin against extensively drug-resistant *Pseudomonas aeruginosa*, including high-risk clones, in an in vitro pharmacodynamic model. *Antimicrob. Agents Chemother.* 2020;64. doi: 10.1128/AAC.02542-19

3. Page 7 paragraph from line 192 - another limitation is the assumptions of PK parameters such as half life and clearance used, which is likely altered in some patients especially the critically ill

- Added.

May 20, 2022

Prof. Maria Milagro Montero
Hospital del Mar. Institut Hospital del Mar d'Investigacions Mèdiques
Infectious diseases
paseo maritim 25-29
barcelona 08003
Spain

Re: Spectrum00892-22R1 (Comparison of ceftolozane/tazobactam infusion regimens in a hollow-fiber infection model against extensively drug-resistant *Pseudomonas aeruginosa* isolates)

Dear Prof. Maria Milagro Montero:

Your manuscript has been accepted, and I am forwarding it to the ASM Journals Department for publication. You will be notified when your proofs are ready to be viewed.

Sincerely,

Aude Ferran
Editor, Microbiology Spectrum
